# The rodent object-in-context task: A systematic review and meta-analysis of important variables

**Milou S. C. Sep**[1,2]☯*, **Marijn Vellinga**[1]☯, **R. Angela Sarabdjitsingh**[1], **Marian Joëls**[1,3]

**1** Department of Translational Neuroscience, UMC Utrecht Brain Center, Utrecht University, Utrecht, The Netherlands, **2** Brain Research and Innovation Centre, Ministry of Defence, Utrecht, The Netherlands, **3** University of Groningen, University Medical Center Groningen, Groningen, The Netherlands

☯ These authors contributed equally to this work.
* m.s.c.sep@umcutrecht.nl

## Abstract

Environmental information plays an important role in remembering events. Information about stable aspects of the environment (here referred to as 'context') and the event are combined by the hippocampal system and stored as context-dependent memory. In rodents (such as rats and mice), context-dependent memory is often investigated with the object-in-context task. However, the implementation and interpretation of this task varies considerably across studies. This variation hampers the comparison between studies and—for those who design a new experiment or carry out pilot experiments–the estimation of whether observed behavior is within the expected range. Also, it is currently unclear which of the variables critically influence the outcome of the task. To address these issues, we carried out a preregistered systematic review (PROSPERO CRD42020191340) and provide an up-to-date overview of the animal-, task-, and protocol-related variations in the object-in-context task for rodents. Using a data-driven explorative meta-analysis we next identified critical factors influencing the outcome of this task, such as sex, testbox size and the delay between the learning trials. Based on these observations we provide recommendations on sex, strain, prior arousal, context (size, walls, shape, etc.) and timing (habituation, learning, and memory phase) to create more consensus in the set-up, procedure, and interpretation of the object-in-context task for rodents. This could contribute to a more robust and evidence-based design in future animal experiments.

## 1. Introduction

Context is defined as a set of independent features that can be observed by an individual and which are stable aspects of the environment [1, 2]. Context-dependent, or contextual, memory is a specific type of episodic memory in which information of events is stored in combination with contextual features [3, 4]. Being able to remember an event with the corresponding contextual information is highly adaptive, since it enables an individual to adjust behavior and respond adequately when encountering a similar event again in a comparable context [2, 5].

**Data Availability Statement:** All data and code files are available from the Open Science Framework (OSF) database (https://osf.io/gy2mc/).

**Funding:** This study was supported by ZonMW grant 'Meer Kennis met Minder Dieren' module

(project #114024150) and the Dutch Ministry of Defense. MS is supported by a personal grant which is part of the Graduate Program (project #022.003.003) of The Netherlands Organization of Scientific Research NWO. MS and MJ were supported by the Consortium on Individual Development (CID), which is funded through the Gravitation program of the Dutch Ministry of Education, Culture, and Science and Netherlands Organization for Scientific Research (project #024.001.003). The funders had no role in study design, data collection and analysis, decision to publish, or preparation of the manuscript.

**Competing interests:** The authors have declared that no competing interests exist.

Conversely, generalizing the response to different contexts may be maladaptive and may contribute to the etiology of psychopathologies, e.g. posttraumatic stress disorder [2], panic disorders [6], phobias [5] or Alzheimer's disease [7].

Context-dependent memory with neutral valence -in contrast to contextual classical or operant conditioning [8]- is in humans often experimentally investigated via the combined presentation of items—like faces [9–12], everyday objects [11–14], or words [15, 16]- and contexts -like scene pictures [9–12, 15, 16], words [14] or sounds [13]- in a computer task [17].

In rodents, the most widely used task to assess neutral context-dependent memory uses physical objects instead of virtual items. This task is commonly known as the object-in-context task (OIC) [18] and relies on the hippocampal system and associated cortical regions [1, 19], comparable to human context-dependent memory [14, 17, 20]. In the OIC test the hippocampal system is needed to establish the link between the object's features and the contextual features; that is, without the hippocampal system an animal is able to remember the object in the sense of familiarity, but not remember the context in which it was encountered [19]. Performance in the OIC task is often used as a behavioral measure of hippocampal function [21]. Moreover, the task is frequently applied to probe context-dependent memory in disease or adversity models, e.g. in animal models of Alzheimer's disease [22], substance (ab)use [23] or early life stress [24].

The OIC task is based on the rodent's spontaneous exploration of objects, which requires (almost) no training [25]. As summarized in Fig 1, the task typically consists of two consecutive phases. In the sample phase the rat freely explores an environment–context A–in which two similar objects are located. This is followed by a second trial in which the rodent is placed in a second environment–context B–with different contextual features and a different set of similar objects. After a certain delay (learning-memory retention time) the test phase is

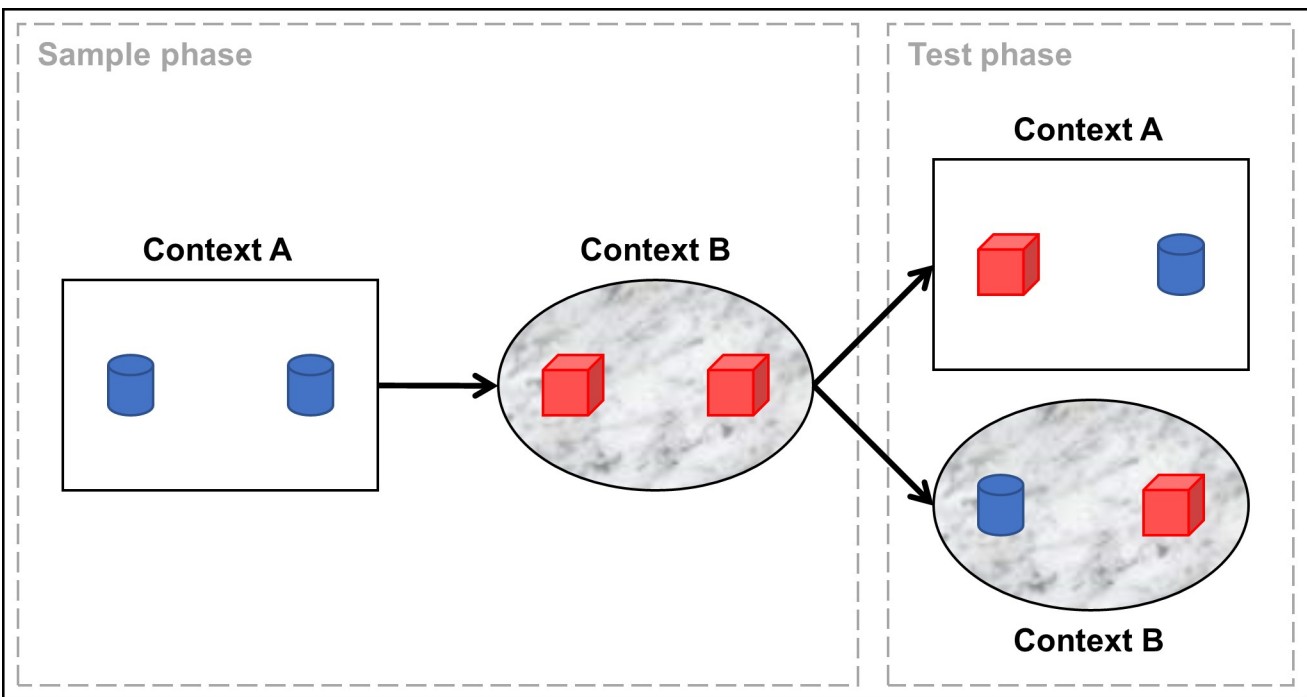

**Fig 1. Schematic overview of the rodent object-in-context task (OIC).** During the sample phase (learning) an animal encounters a unique set of two objects in context A and next a different set in B. In the test phase (memory) the animal is exposed to either context A or B, with one object of each unique set.

conducted, in which the rat is placed in either context A or B, but this time with one object previously encountered in context A and one object previously encountered in context B. As a result, one object is new in that environment (i.e. novel), while the other has been encountered before in the same environment (i.e. familiar). In the OIC task, contextual information is needed to discriminate between objects. According to the so-called novelty preference paradigm [26] it is assumed that rodents have a preference for novel over familiar objects. Animals will explore the novel object more often if they remember the object-context combinations from the sample phase, reflecting their context-dependent memory. The main outcome measure of the OIC task is the discrimination ratio (DR), which is calculated for each animal. This is based on the time the animal spends exploring the novel object in relation to the time spent exploring the familiar object and the total exploration time.

Despite these general principles of the task, there is a great deal of variation possible in the set-up, procedures and interpretation. Variations in set-up and procedure might have consequences for the animals' behavioral performance and hinder direct comparison between studies. To chart these variations and particularly to identify critical factors determining experimental outcome, we performed a systematic literature review of studies that employed the OIC task in control animals, i.e. naive, sham-operated or saline-injected rodents. As a follow-up, we determined the average DR for control animals using a meta-analytical approach. Next, a data-driven exploration was used to identify which variations affect the behavioral outcomes of this task. Considerable variation in OIC implementation among published studies is expected, and we expect that (some of) these variations affect animals' performance. In the Discussion, we integrated the observations into methodological recommendations for future studies using the OIC task and provide a critical reflection on the (novelty preference) assumptions of the OIC.

## 2. Methods

This study was performed and reported in accordance with the SYRCLE [27, 28], PRISMA [29] and ARRIVE guidelines [30], and preregistered in PROSPERO (CRD42020191340) [31]. The PRISMA checklist is provided in A10 Appendix in S1 File. The documents, datasets and code used during literature search, screening, data extraction, and meta-analyses are available via Open Science Framework (OSF; https://osf.io/gy2mc/). The collected dataset allows for the development of web-based explorative tools, that can become available on the OSF webpage.

### 2.1. Search strategy

A comprehensive literature search was performed in the electronic database PubMed. The search string contained search terms for '*learning and memory*', '*rodents*', '*object-in-context task*' and terms to exclude *meta-analyses* and *systematic reviews* (the complete search filter is provided in A1 Appendix in S1 File). The final search was performed on 25[th] of May 2020. References of included publications were checked for eligibility (snowballing).

### 2.2. Screening

The retrieved articles were screened based on *a priori* defined inclusion and exclusion criteria. Inclusion criteria for the systematic review were: (1) original article in the English language, (2) OIC task, (3) rodents, (4) control animals (no treatment, saline injection, or sham operation). Studies had to comply with the inclusion criteria above and report the sample size and (the data to calculate) the DR-with the corresponding standard error- to be included in the meta-analysis. Exclusion criteria were: 1) no primary literature in English; 2) other measures of context-dependent memory, like modified versions of the OIC task (e.g. combinations of

context and location measures, context-dependent memory based on odors, classical fear conditioning or operant conditioning paradigms); 3) non-rodent species, 4) no control group tested. Note, genetic modification was not an exclusion criterium. Screening was performed by MV and MS. If information in the title and abstract was insufficient to determine eligibility, full-text articles were checked.

## 2.3. Data extraction and study quality assessment

Extracted data was *a priori* defined. The complete data-extraction codebook is provided in A2 Appendix in S1 File, A1 Table in S1 File, and includes: 1) Publication details (authors, year of publication, etc.); 2) methodological details, for instance context-dependent differences between boxes, object characteristics, habituation protocol, trail duration, retention time between phases, order in which contexts are encountered, behavioral scoring, etc.; 3) animal characteristics such as species, strain, sex, age, previous use in experiments, type of control group, housing conditions (e.g. day-night cycle, group-housing) etc. In addition, sample size and time spent exploring the objects to calculate the DR (with standard error) were extracted for studies included in the meta-analysis. Note, the mean sample size was calculated if only a range was provided. Plot Digitizer [32] was used to extract visually presented data from graphs. Authors were not contacted for missing or additional data, missing values were included in dataset and further processed as described in section 2.4.2. Initial data-extraction was performed by MV, 20% of the studies were independently checked by MS.

As multiple formulas to calculate DR for object recognition are described in literature [33], all extracted DRs were transformed to center around 0 (i.e. DR = (novel–familiar) / (novel + familiar)). The corresponding standard errors of transformed DRs were recalculated accordingly. For transformation formulas, see A3 Appendix in S1 File.

Study quality and the risk of bias were assessed with SYRCLE's risk of bias tool [34] by MV. Unreported details were scored as an *unclear* risk of bias. MS independently scored 10% of the studies, discrepancies were discussed until consensus was reached.

## 2.4. Statistical analysis

Statistical analyses were based on earlier work of our group [35] and performed with $\alpha$ = .05 in R version 4.0.3 [36], with the use of packages *dplyr* [37], *osfr* [38], *metafor* [39], *metaforest* [40], *caret* [41] and *ggplot2* [42].

**2.4.1 Random-effects meta-analysis: Estimation of overall effect.** The (transformed) observed mean DR was used as effect size, i.e. raw mean. Sampling variance was calculated from the (transformed)observed standard error and sample size. As heterogeneity in the data was expected, the overall effect size (i.e. overall mean DR) was estimated with a nested random effects model with restricted maximum likelihood estimation [43]. The estimation was nested within *articles* and *experiments*. Heterogeneity was assessed with Cochrane Q-test [39] and the $I^2$-statistic (low: 25%, moderate: 50%, high: 75% [44]). Robustness of the estimated effect was evaluated via *Rosenthal's fail-safe N* [45] and trim-and-fill analyses [46]. Moreover, funnel plot asymmetry -as index for publication bias- was tested via *Egger's regression* [47] *and Begg's test* [48]. Sensitivity analyses were performed to evaluate if the estimated effect was influenced by 1) study quality and/or 2) influential cases or outliers [49]. To evaluate the influence of study quality, the scores on SYRCLE's risk of bias tool for randomization (0–5), blinding (0–2) and reporting (0–2) were combined into a summary quality score (formula in the A4 Appendix in S1 File).

**2.4.2 Random forest for meta-analysis: Exploration of heterogeneity.** To explore the source heterogeneity (variation between studies), a random forest-based meta-analysis was

performed using *MetaForest* [40]. This data-driven approach allows to rank potential moderators of the overall effect, based on variable importance in the random forest.

Variables with more than 1/3 missing values, indicated in A2 and A7 Appendices in S1 File, were excluded from the random forest analysis. The missing values in other variables were replaced by median value (for continuous variables) or most prevalent category (for categorical variables).

As certain variables are part of one underlying factor, four summary scores were added to the analysis (formulas are provided in the A5 Appendix in S1 File): one context difference score and three arousal scores: 1) arousal prior to OIC, 2) arousal related to OIC habituation procedures and 3) their combination. Of note, no cumulative object difference score was created for object material and size, as object size contained more than 1/3 missing values and was excluded from analyses (note, the object material variable was included in the analyses).

The random forest-based meta-analysis (600 trees) was tuned (optimal parameters for minimal RMSE: uniform weighting, 4 candidate moderators at each split, and a minimum node size of 2) and 9-fold cross-validated.

To follow-up the most important moderators, i.e. the upper 50% based on random forest variable importance, partial dependence (PD) plots were used to explore the relations between moderator and DR. PD plots show the predicted effect at different levels of a particular moderator, if all other moderators are kept constant [40, 50]. In addition, weighted scatter plots were created to inspect the distribution of the raw data per moderator.

## 3. Results

### 3.1. Study selection and characteristics

After screening 254 unique studies (one publication was identified via snowballing), 41 articles were included in the systematic review. Four of these did not report on sample size, standard error or standard deviation and had to be excluded from the meta-analyses. In total 37 papers with 857 unique animals were included in the analyses. The flowchart is shown in Fig 2 and the study characteristics are provided in A6 Appendix in S1 File (A2 Table in S1 File).

The results of the systematic review are shown in A7 Appendix in S1 File (A3 Table in S1 File), and reveal extensive variation in animal characteristics, set-up and task procedures' related factors. Although all rodent species were eligible, the screening process only returned rat and mice papers. Study's quality is shown per item in Fig 3 and the cumulative study scores are shown in A8 Appendix in S1 File (A1 Fig in S1 File). Many studies did not report on all potential risks of bias (Fig 3): only 6 out of 41 articles (~15%) reported that 4 or more -out of SYRCLE's 9 [34]- measures were taken on randomization, blinding and / or reporting to reduce the risk of bias (A1 Fig in S1 File).

### 3.2. Meta-analysis of Discrimination Ratio (DR)

We next performed a meta-analysis on the available data (forest plot in Fig 4). This revealed a high degree of heterogeneity in the multilevel model ($Q(80) = 406.307$, $p < .0001$; $I^2 = 82.994$); 77% due to variance between studies and 6% due to variance within studies. The overall estimated DR was significantly different from 0 (mean DR = 0.2579, SE = 0.0266, 95% CI = 0.2057–0.3101, z = 9.6879, p < .0001), indicating that animals discriminated significantly between the in- and out-of-context objects. The standard deviation of the estimated overall DR is 0.7785 (SD = SE * $\sqrt{\text{n}}$ unique animals in meta-analysis [51] = 0.0266 * $\sqrt{856.5}$) and the estimated effect size of the difference between the estimated overall DR and 0 is small to

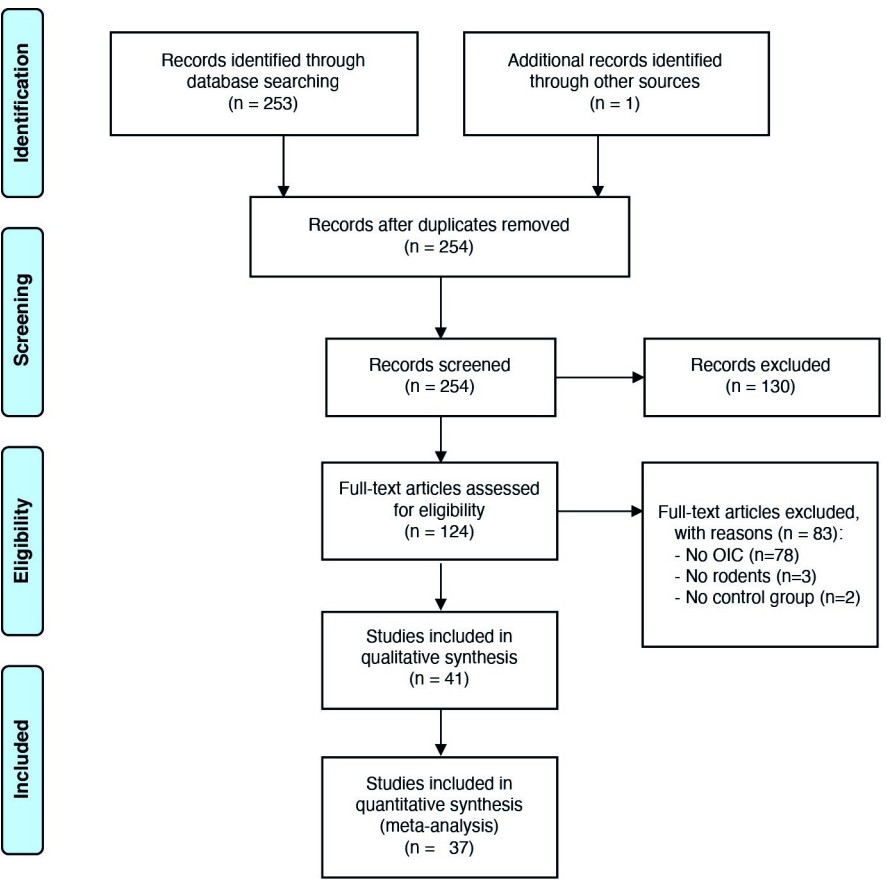

**Fig 2. Flowchart of the study.**

medium: Cohen's d = (μ—μ0) / SD = (0.2579–0) / 0.7785 = 0.3313 [52]. Given this effect size, 58 animals would be required to detect a significant difference from 0 in a control group (G*Power [53], one-side t-test, α = 0.05, power = 0.80).

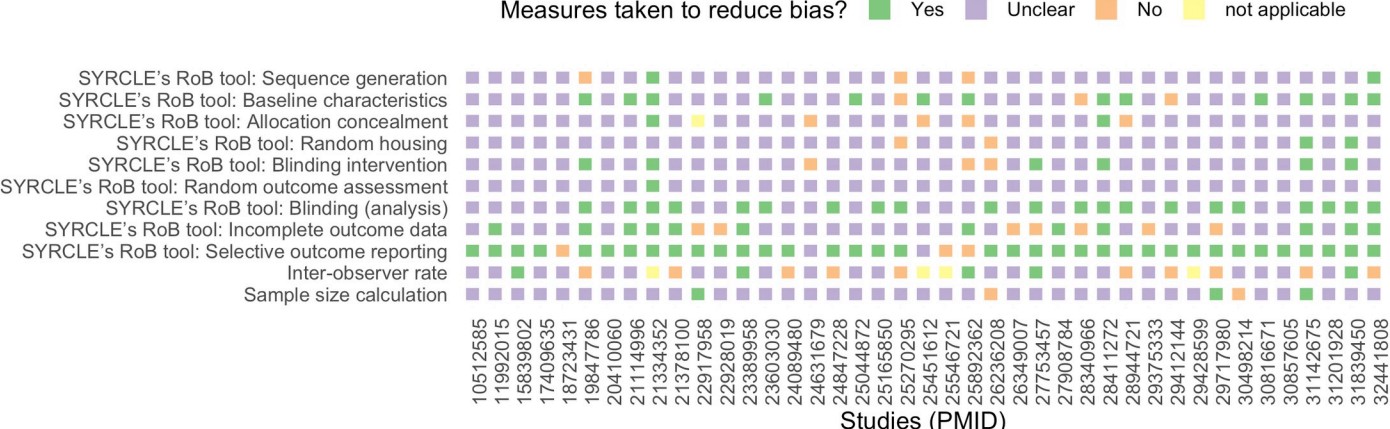

**Fig 3. Assessment of study quality (QA) and risk of bias.** The risk of bias according to SYRCLE's risk of bias (RoB) tool [34], for each study, indicated by PubMed ID (PMID), included in the systematic review. The figure also shows if an a priori sample size calculation was performed and -if behavior was scored manually- an inter-observer rate was calculated. Unreported details were scored as an 'unclear' risk of bias.

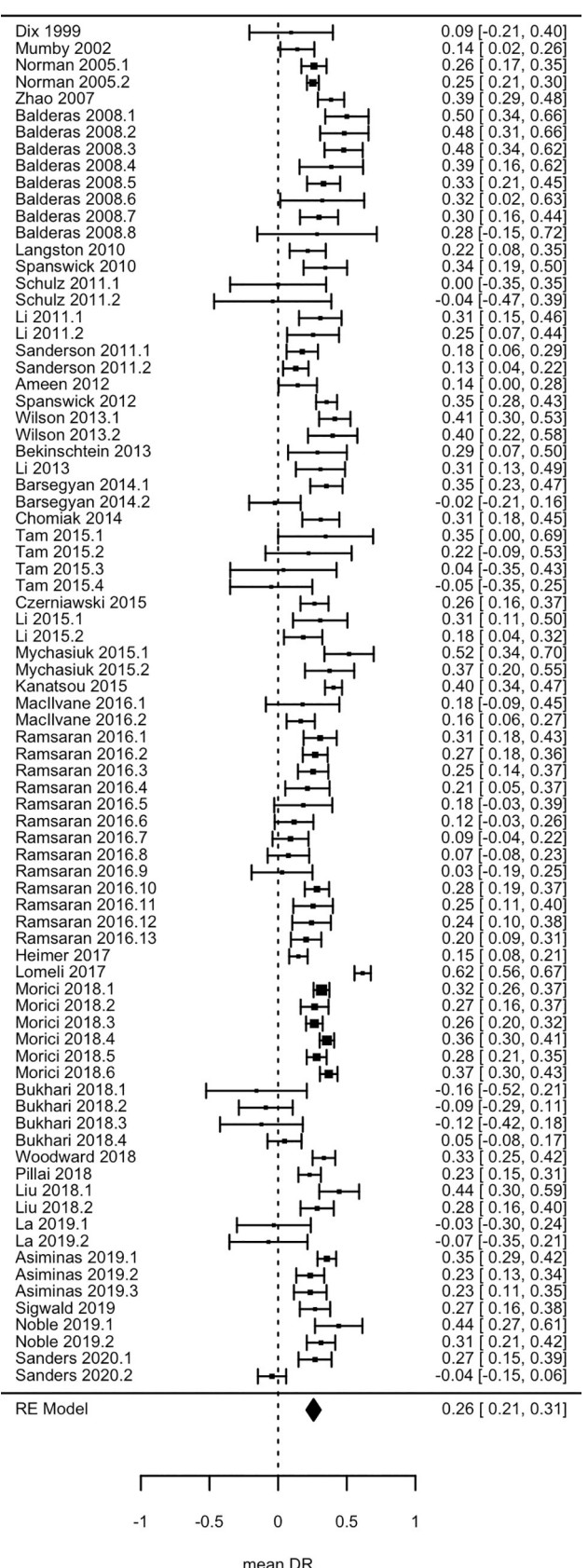

**Fig 4. Forest plot.** Forest plot visualizing the Discrimination Ratio (DR) per experimental group, per study; and the overall mean DR with 95% Confidence Interval. Studies are presented by publication year in ascending order.

## 3.3. Robustness of the estimated DR: Sensitivity analyses and publication bias

The presence of publications bias is suggested by qualitative examination of funnel plot asymmetry (Fig 5), and was confirmed by Egger's regression (z = -3.492, p < .001) and Begg's test (z = -3.4925, p = .0005). However, file drawer analyses suggested that the influence of publication bias was limited: 12 (SE = 5.8941) studies were missing on the right side, according to trim-and-fill analysis, and 43154 studies would be needed to nullify the estimated effect, according to Rosenthal's fail-safe N analysis.

Furthermore, sensitivity analyses revealed that 1) study quality -assessed with SYRCLE's RoB tool- did not moderate the estimated overall DR (QM(1) = 1.9511, p = 0.1625); 2) there were no potential influential cases; and 3) there were four potential outliers, though without substantial impact on the results, i.e. their exclusion led to a similar estimated overall DR: mean DR [95%CI] = 0. 2599 [0.2165,0.3033], z = 11.7398, p < .0001).

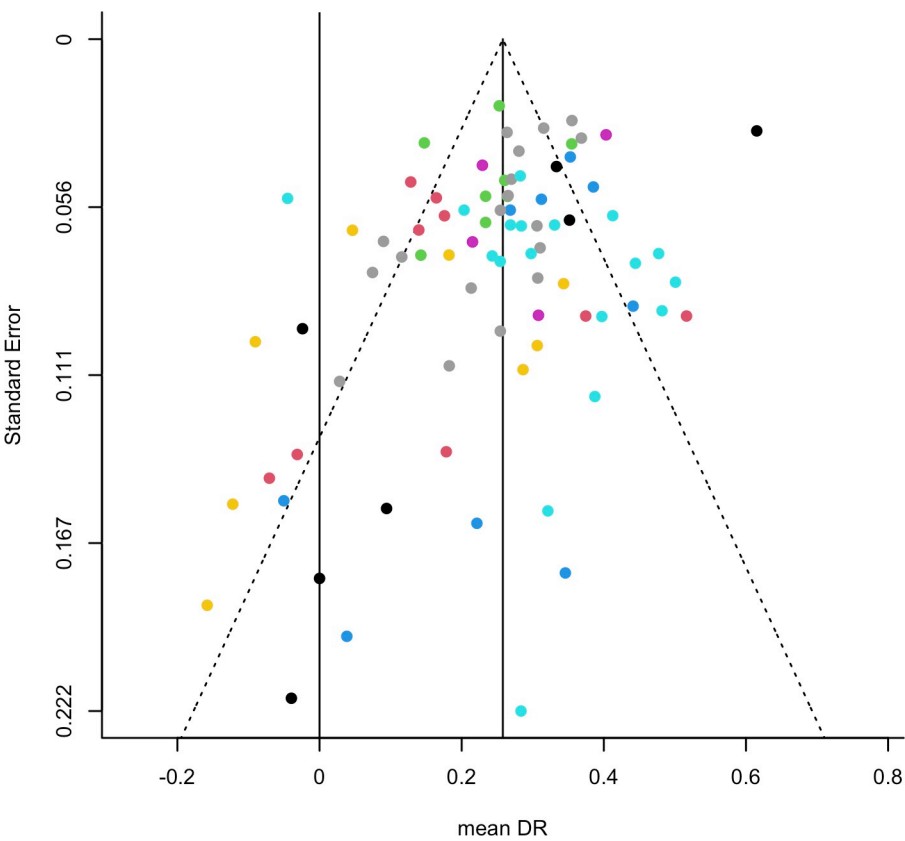

**Fig 5. Funnel plot.** Funnel plot showing the DR of individual control groups on the x-axis against their standard errors (i.e., the square root of the sampling variances) on the y-axis. Vertical reference lines indicate the 0, i.e. no context-dependent memory; and the estimated overall mean DR based on the model. Colors indicate unique studies (PMID).

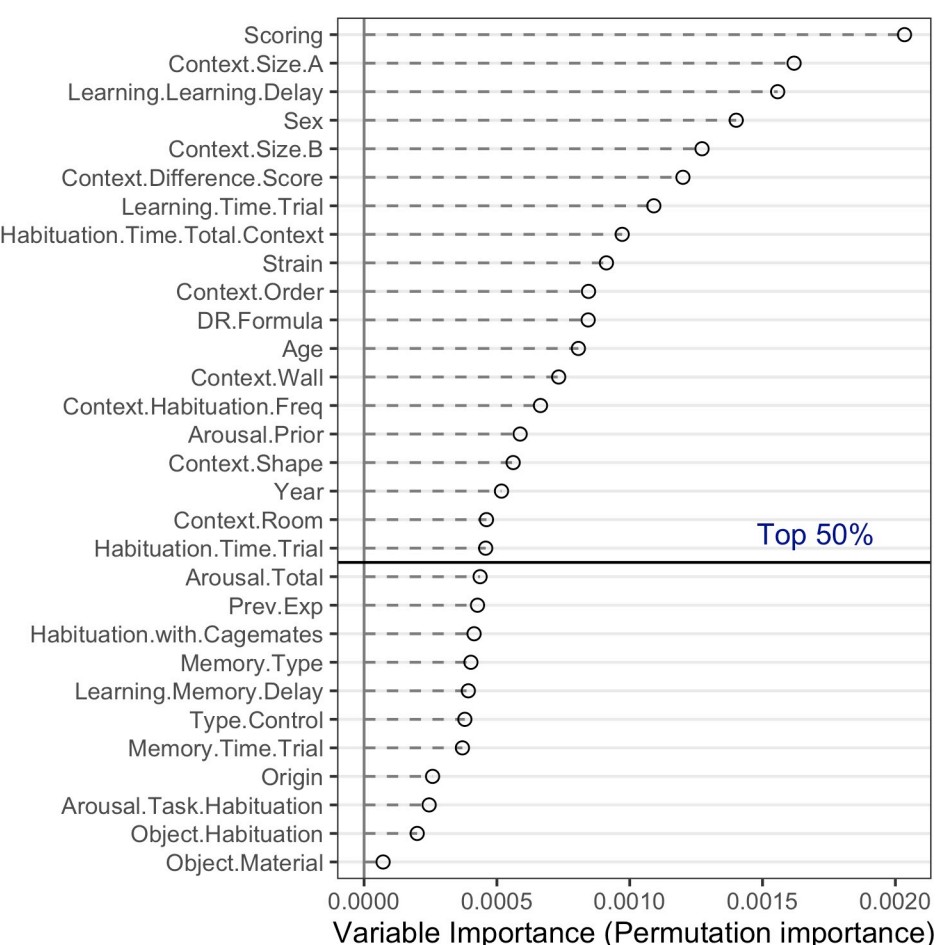

**Fig 6. Relative importance of potential moderators based on 'permuted variable importance' in the random forest-based meta-analyses.** Moderators above the horizontal line belong to the top 50%. Full definitions of variables and sum scores can be found in A2 and A5 Appendices in S1 File respectively.

## 3.4. Exploration of moderators: Random-Forest variable importance

The data-driven exploratory random forest-based meta-analysis provided insight in the sources of heterogeneity in the sample. Our 9-fold cross-validated random forest model showed good convergence (see A9 Appendix in S1 File, A2 Fig in S1 File) and the included moderators accounted for 37.5% of the variance ($Rcv^2$[SD] = 0.375[0.215]). The ranking of potential moderators–based on permuted random forest variable importance- is shown in Fig 6.

The upper 50% of the most important variables included four animal-related factors (Sex, Strain, Age, Arousal.Prior), six set-up related factors (Context.Size.A, Context.Size.B, Context. Difference.Score, Context.Wall, Context.Shape, Context.Room), and seven procedure-related factors (Scoring, Learning.Learning.Delay, Learning.Time.Trial, Habituation.Time.Total.Context, Context.Order, Context.Habituation.Freq, Habituation.Time.Trial) as well as two other factors (DR.formula and Year). Note, precise variable and sum score definitions are provided in A2 and A5 Appendices in S1 File respectively.

Exploratory partial dependence plots show the predicted relations of these top 50% selected moderators and the DR in the OIC task, for rats and mice separately (Fig 7). Considering the top-5 most influential factors in both rats and mice, it was evident that DR was higher in males

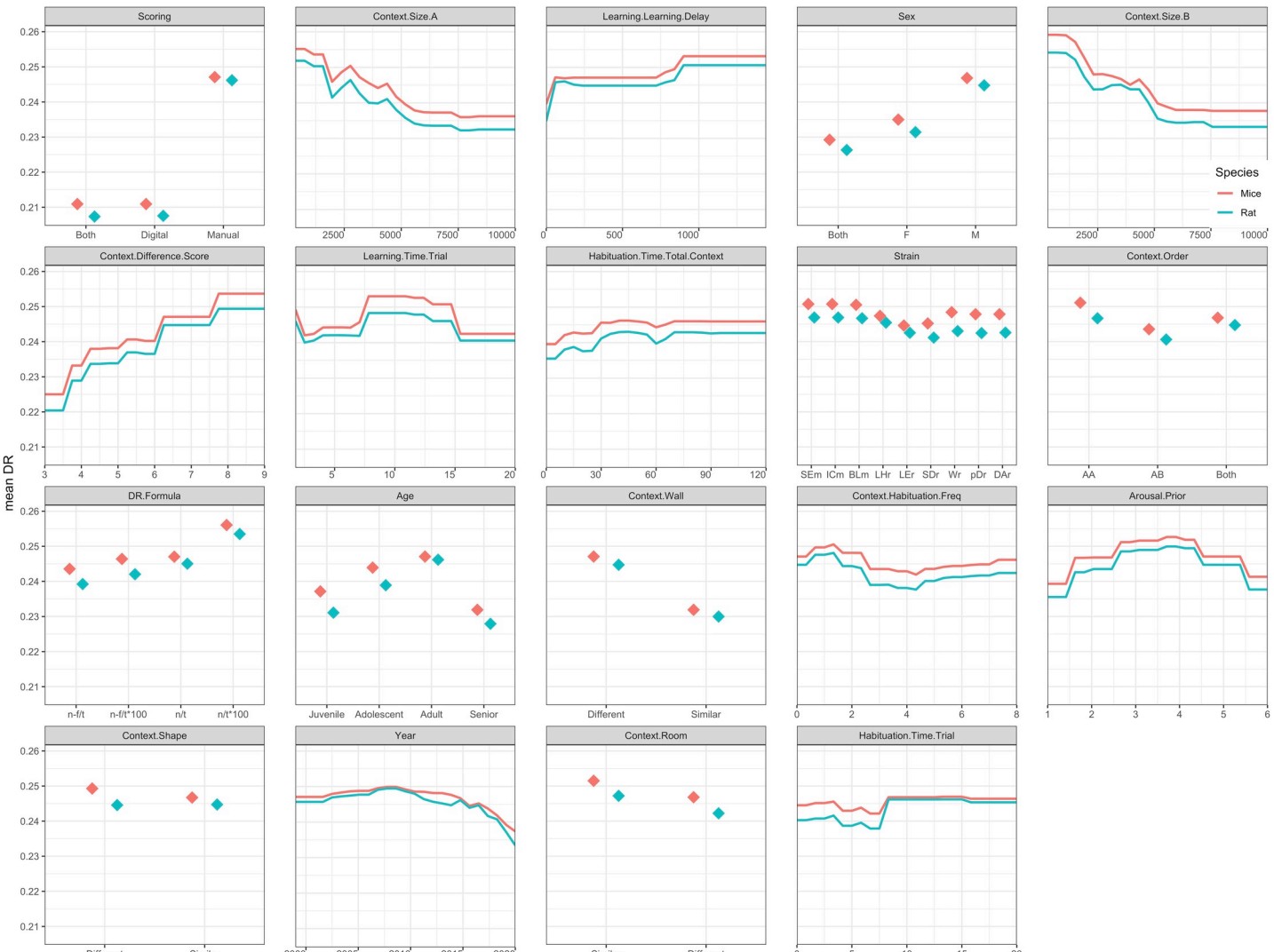

**Fig 7. Partial dependence plots showing the predicted relation between DR and the upper half most important variables in the random forest-based meta-analysis, broken down by species: Mice in red vs. rat in blue.** The y-axis shows the predicted DR and x-axis shows the values of the variable that is named above the graph (in gray). The values of context.size A and B are shown in in $cm^2$ and the values of time variables (learning. learning.delay, learning.time.trial, habituation.time.total.context, habituation.time.trial) are shown in minutes. Higher context.difference.score values indicate more different contexts and higher arousal.prior values indicate more arousal prior to the experiment. Strains values are abbreviations: TMm (Tg(Sim1cre)KH21Gsat/Mmucd mice), SEm (SEm Sv/Ev mice), ICm (ICR mice), BLm (C57BL/6 mice), LHr (Lister hooded rats), LEr (Long-Evans rats), SDr (Sprague-Dawley rats), Wr (Wistar rats), pDr (pigmented DA strain rats), DAr (Dark Agouti rats). DR. formula values are also abbreviations: n-f/t ($(T_{novel}-T_{familiar})/(T_{novel}+T_{familiar})$), n-f/t*100 ($[(T_{novel}-T_{familiar})/(T_{novel}+T_{familiar})]*100$), n/t ($(T_{novel})/(T_{novel}+T_{familiar})$), n/t*100 ($[(T_{novel})/(T_{novel}+T_{familiar})]*100$). The context.order value AA indicates that memory was tested in the context of the second learning trial, the value AB indicates that memory was tested in the context of the first learning trial, 'both' indicates that AA and AB were randomized. Finally, context.habituation.freq values show the number of visits per context. Full definitions of variables and sum scores can be found in A2 and A5 Appendices in S1 File respectively. Note, predictions based on the random forest-model do not take into account that strains belong to a specific species (either rat or mouse), hence rat and mice predictions were generated of all strains.

(versus females or mixed populations); when animals were tested in relatively small (< 2500 $cm^2$) context boxes (A and B); and when the delay between the two learning sessions was not too short (> 825 min). Also, the method of scoring (manual versus digital) was found to be of influence: manual scoring led to a higher DR.

Weighted scatter plots showing the distribution of the raw DR by the top 50% selected moderators are presented in a Fig 8. From the top-5 most influential factors, the most variation in DR between animals was observed for males (versus females or mixed groups); when larger

context boxes were used (~ 2500 cm$^2$); and when a very brief delay between learning trials was applied. Variation in the DR was also affected by the scoring method (manual vs digital).

## 4. Discussion

Object-in-context learning is frequently used to probe contextual memory formation and (dorsal) hippocampal function in rodent models of disease or adversity [54]. However,

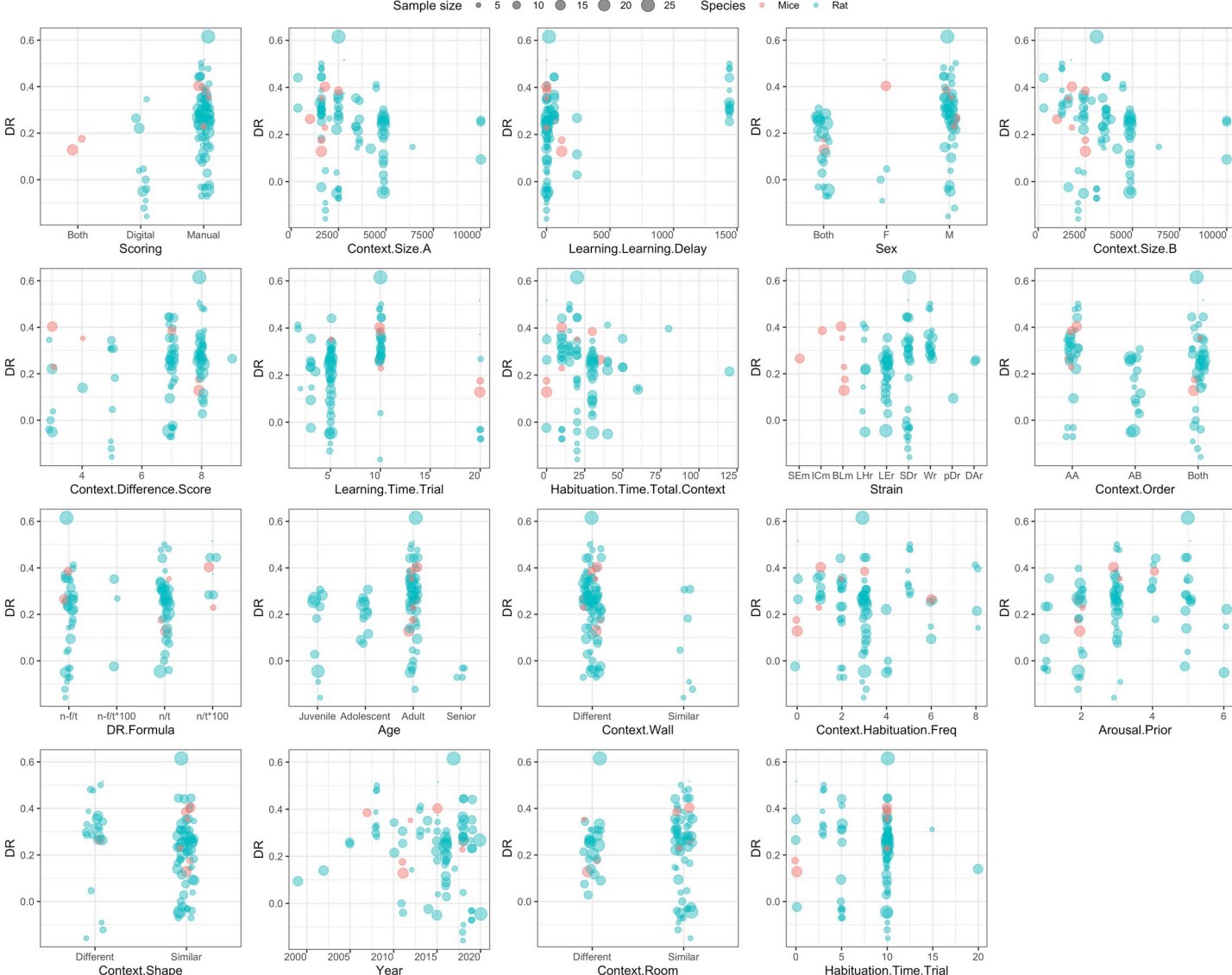

**Fig 8. Weighted scatter plots showing the distribution of the raw DR for the 50% most important variables in the random forest-based meta-analysis, broken down by species: Mice in red vs. rat in blue.** Dot size indicates the sample size of each observation. Most studies were performed in rats. The y-axis shows the mean DR and x-axis shows the values of the variable that is named below the graph. The values of context.size A and B are shown in in cm$^2$ and the values of time variables (learning. learning.delay, learning.time.trial, habituation.time.total.context, habituation.time.trial) are shown in minutes. Higher context.difference.score values indicate more different contexts and higher arousal.prior values indicate more arousal prior to the experiment. Strains values are abbreviations: TMm (Tg(Sim1cre)KH21Gsat/Mmucd mice), SEm (SEm Sv/Ev mice), ICm (ICR mice), BLm (C57BL/6 mice), LHr (Lister hooded rats), LEr (Long-Evans rats), SDr (Sprague-Dawley rats), Wr (Wistar rats), pDr (pigmented DA strain rats), DAr (Dark Agouti rats). DR.formula values are also abbreviations: n-f/t (($T_{novel}$-$T_{familiar}$)/($T_{novel}$+$T_{familiar}$)), n-f/t*100 ([($T_{novel}$-$T_{familiar}$)/($T_{novel}$+$T_{familiar}$)]*100), n/t (($T_{novel}$)/($T_{novel}$+$T_{familiar}$)), n/t*100 ([($T_{novel}$)/($T_{novel}$+$T_{familiar}$)]*100). The context.order value AA indicates that memory was tested in the context of the second learning trial, the value AB indicates that memory was tested in the context of the first learning trial, 'both' indicates that AA and AB were randomized. Finally, context.habituation.freq values show the number of visits per context. Full definitions of variables and sum scores can be found in A2 and A5 Appendices in S1 File respectively.

variations in animal-, set-up or procedure-related factors seriously hamper comparison between studies. This was confirmed by the current systematic review and subsequent meta-analysis of 37 studies that employed the OIC task in (control) rats and mice. Overall, the DR differed significant from 0, indicating that rodents on average do discriminate between the in- and out-of-context objects, reflecting the formation of context-dependent memory. Yet, this effect was small to medium and accompanied by a large degree of heterogeneity, mostly (77%) due to variance between studies. As expected, a substantial part (37.5%) of the variance could be explained by a set of moderators identified by a random forest approach, with prominent roles for e.g. sex, size of the boxes and delay between the learning trials. In the following sections we will first discuss methodological considerations of our approach and next provide recommendations regarding the set-up, procedure and interpretation of the OIC task in rodents for future users.

## 4.1. Methodological strengths and limitations

Based on our samples size (37 studies), robust findings can be expected. The quality of meta-analyses, however, depends on the quality of the studies on which these analyses are based. The fact that only ~15% of all studies reported 4 or more out of SYRCLE's 9 measures to reduce risk of bias [34] might suggest poor study quality. However, study quality did not moderate the estimated overall DR. Moreover, as this percentage was influenced by unreported details in many studies, it is probably an underestimation, since practices like randomization or blinding may have been applied but simply not reported. Still, lack of reporting introduces an unclear risk and at least difficulty to estimate the quality of the studies [55]. The problem of insufficient reporting of experimental details and quality measures in pre-clinical studies has been extensively addressed and previously observed in meta-analysis [35, 56–60].

Since our analyses were based on metadata, we cannot exclude that studies were liable to p-hacking practices, such as post-analysis decisions which variables to report on, whether or not to include outliers or stopping data exploration once a significant *p*-value was reached [61]. In that case one might expect to see publication bias, for which we found only suggestive evidence in the funnel plot. Subsequent sensitivity analyses did not confirm this. Also, potential outliers among the studies did not affect the overall outcome. All in all, assessments of study quality did not indicate severe limitations in the use of the current dataset.

We adopted the state-of-the-art random forest-based meta-analysis *MetaForest* [40] for an unbiased exploration of the sources of heterogeneity in our dataset. This technique is robust to overfitting, can identify non-linear relationships and is valid for meta-analysis with 20 or more studies [62]. Relative variable importance, derived from the random forest model, was used to identify the most important moderators of the DR in control animals. Not all extracted variables could be included in these explorative analyses, due to too much missing values. As we could not judge the potential importance of these excluded variables, the provided overview might be incomplete.

Partial dependance plots were used to visually inspect the marginal effects of these selected moderators on the DR [62]. Although these plots are the most widely used to explore relations in black box models like random forests [50], the visualized relations need to be interpreted with some caution, especially when based on few or unequally distributed raw observations (e.g. *learning.learning.delay*).

## 4.2. Rethinking the DR definition

The main outcome measure of the OIC task is the discrimination ratio (DR). Two ways have been used to calculate this ratio: Either $DR = T_{novel} / T_{total}$ [63] or $DR = (T_{novel} - T_{familiar}) / T_{total}$ [18].

In our analysis, all extracted DRs were recalculated to center around 0 (i.e. DR = ($T_{novel}$ − $T_{familiar}$) / $T_{(novel + familiar)}$). Interestingly, the average DR was 0.26 and the estimated effect size of its difference to 0 was small to medium. To detect this effect, future studies would require a sample size of 58 animals per group. As such sample sizes are seldomly seen in pre-clinical control groups, future studies could benefit from the use of historical data on the OIC task to increase statistical power with a limited sample size [64]. Since the OIC task is often used to examine shifts in contextual memory in animal models for disease [22, 23] or (early life) adversity [24], such low values in controls might introduce a 'floor effect', i.e. a lower DR in the experimental group might remain undetected given the already low DR in the controls.

Of particular interest is the interpretation of the DR. It is generally assumed [26] that rodents have a preference for novel objects over familiar objects and thus will explore the novel object more often if they remember the object-context combinations from the sample phase. However, one could also argue that consistent preference for the *familiar* object in a context may reflect context-dependent memory, assuming a more 'conservative strategy' in which e.g. neophobia or lack of boldness outweigh innate curiosity (reviewed in [65]). If so, DRs that differ considerably from 0 (assuming transfer around 0, with 0 indicating performance at chance level) would reflect the formation of a contextual memory, regardless of the sign. To rule out this influence of differences in strategy (the latter being a composite of many underlying factors like curiosity, boldness or neophobia), one would then prefer to use the absolute value of the difference between time spent with the novel and familiar object in a context, as a function of the total object exploration time [66]. The proposed "absolute DR" [66] is preferable when the aim is to measure (experimental influences on) context-dependent memory, regardless of the animal's strategy. However, when the aim is to examine (experimental influences on) an animal's strategy, the sign of the DR (centered around 0) holds valuable information.

## 4.3. Critical factors contributing to the DR in control animals

Based on the unbiased random forest approach, we conclude that many variables affect the DR in the OIC task. Together, these variables explain 37.5% of the variation between and within studies.

Regarding the animal-related factors, it was interesting that very little difference was observed between studies with rats versus mice, although it should be noted that only few studies (<10%) were carried out with mice, so that any conclusions about mice should be made with care. Among the studies with rats, clear strain differences were observed: Sprague Dawley or Long Evans rats displayed much more variation than e.g. Wistars. Interestingly, these strain differences align with earlier observations in another visuo-spatial learning task [67]. In terms of age, highest DR values were found in adulthood, with on average lower values in younger or older animals. This is of interest, since context-dependent memory in humans shows a similar inverted U-shaped age-dependency [68]. Memory context-dependency increases with age in children, as they develop the ability to bind and integrate information [69]. As adults get older, context dependency decreases, which has been linked to age-related reductions in selective attention -leading to hyper-binding of too much contextual details thereby reducing accuracy for the relevant context- [70, 71]; alterations in prefrontal-hippocampal connectivity [72]; and declines in hippocampal neurogenesis [73, 74]. In line with frequently reported sex-dependence of spatial or neutral contextual memory formation across species [75, 76], male animals showed a higher DR than females, but firm conclusions await more studies in females. Performance in the OIC might be enhanced in males compared to females, as the former are more likely to adopt a hippocampus-dependent place-strategy to navigate an environment (i.e.

navigate based on a contextual map) than the latter (who are more likely to use a striatum-dependent landmark strategy) [77]. Such place-strategies could create a stronger representation of the OIC context in the brain, enabling richer context-dependent memories. Finally, an often-neglected factor concerns (potential) exposure of animals to arousal prior to being tested. As with age, an inverted U-shaped dependency was observed for prior arousal, with highest DR values in moderately aroused animals. Inverted U-shaped dose-dependency is a common phenomenon in stress-related influences on memory formation [78]. Of note, it is also known that arousal and stress can affect memory formation and retrieval differently [79]. As a consequence, the time of saline-injection -which can trigger a mild stress response [80]-with respect to the OIC phases (e.g. before sample or test phase; or exactly when relatively to the test phase) could have caused variation in the saline-injected control animals, which was not accounted for in the current meta-analysis. However, type of control group ranked among the less important variables (bottom 50%), suggesting that variation from this source might have limited effects on overall performance.

Next to animal-related factors, factors related to the experimental set-up or learning paradigm also influenced OIC outcome. In general, it transpired that variation in context affected the outcome much more strongly than object (material) variation. Smaller arenas resulted in higher DR values, as did large difference between context A and B. Recency effects play a role in the DR values, as higher DR values were observed when memory was tested in the context that was used in the last (second) learning trial, as opposed to the first learning trial. Interestingly, higher DR values were also observed when the delay between learning trials was longer, while a shorter delay let to more individual variation in the DR.

Finally, one of the factors contributing most clearly to the variation was the way of scoring: DR values were found to be much higher in studies using manual scoring than in those using automated scoring programs. It cannot be excluded that this was somewhat affected by (the absence of) blinding of the experimenter to groups and/or object type (novel vs familiar). Manual scoring without proper blinding could lead to more subjective interpretation of an animal's interaction with one of the objects, whereas automated scoring is generally combined with blinding of the experimenter, which leads to more trustworthy results.

## 4.4. Recommendations for future studies

Based on the random forest analysis, recommendations for optimizing DR values are summarized in Table 1. We considered two angles in the design: if one is primarily interested in studying context-dependent memory formation, the task should be designed such that the (absolute) DR value is as high as possible, with very little variation between animals in the control group. However, if one is also interested in individual differences in context-dependent memory formation, variation over the entire spectrum of (absolute) DR values is welcome. Of note, the factors indicated in Table 1 are currently based on qualitative rather than quantitative interpretation. The exact degree to which they contribute to the overall outcome of the test was not determined in the current analysis, which is a limitation of the study.

All in all, the current study illustrates that insights from historical datasets can help to interpret data from control animals, which can next be used to increase power of future studies [64]. Performing an unbiased data-driven analysis of metadata may form the basis for more consensus on the set-up, procedure and interpretation of the OIC task for rodents; and hence for recommendations how to design future studies. This may be particularly helpful for those who have never used the task before. But even for more experienced investigators, awareness of factors influencing the dependent variable may help to optimize the experimental design.

**Table 1. Recommendations for future animal studies.**

| | | |
|---|---|---|
| **DR calculation:** | $\Rightarrow$ Aim to study context-dependent memory: absolute DR = abs($T_{novel} - T_{familiar}$) / $T_{total}$ | |
| | $\Rightarrow$ Aim to study animals' strategy: the sign (positive vs negative) of DR as calculated with ($T_{novel} - T_{familiar}$) / $T_{total}$ | |
| **Factors:** | **1) for the highest mean DR** *(based on PD plots in Fig 7:* [1] *Sex;* [2] *Strain;* [3] *Age;* [4] *Arousal.Prior;* [5] *context.size.B & context.size.A;* [6] *context.difference.score;* [7] *context.wall;* [8] *context.shape;* [9] *context.room;* [10] *Habituation.time.total.context;* [11] *context.habituation.freq;* [12] *habituation.time.trial;* [13] *learning.learning.delay;* [14] *Learning.Time.Trial;* [15] *context.order)* | **2) for the most individual variation in DR** *(based on WS plots in Fig 8:* [16] *Sex;* [17] *Strain;* [18] *Age;* [19] *context.size.B & context.size.A;* [20] *context.wall;* [21] *context.shape;* [22] *context.room;* [23] *context.habituation.freq;* [24] *habituation.time.trial;* [25] *learning.learning.delay;* [26] *Learning.Time.Trial;* [27] *context.order)* |
| Animal related factors | • Males [1]<br>• Not Long-Evans or Sprague-Dawley rats [2]<br>• Adults [3]<br>• Medium levels of arousal prior to testing [4] | • Males [16]<br>• Long-Evans or Sprague-Dawley rats [17]<br>• Adults [18] |
| Set-up related factors | • Context boxes < 2500 cm$^2$ [5]<br>• Most different contexts [6], especially:<br> ○ Different walls [7]<br> ○ Different shape [8]<br> ○ Different room [9] | • Context boxes ~2500 cm$^2$ [19]<br>• Different context walls [20]<br>• Similar context shape [21]<br>• Similar context room [22] |
| Task procedure related factors | <u>Habituation:</u><br>• Total habituation time per context > 30 minutes [10]<br>• 2 or less habituation trials per context [11]<br>• > 7.5 minutes habituation per trial [12]<br><br><u>Learning (sample phase):</u><br>• Delay between learning trials > 825 min [13]<br>• Time of learning trials between 7.5–15 minutes [14]<br><br><u>Memory (test phase):</u><br>• Testing in last learning context [15] | <u>Habituation:</u><br>• 3 habituation trials per context [23]<br>• 10 minutes habituation per trial [24]<br><br><u>Learning (sample phase):</u><br>• Brief delay between learning trials [25]<br>• 5 minutes learning trials [26]<br><br><u>Memory (test) phase:</u><br>• Counterbalanced testing in first and last learning context [27] |

## Supporting information

**S1 Checklist. Prisma checklist.**
(PDF)

**S1 File. Appendix.**
(PDF)

## Acknowledgments

We are very thankful to Valeria Bonapersona for here valuable input on the statistical analyses. We thank Elbert Geuze for the opportunity to conduct this research.

## Author Contributions

**Conceptualization:** Milou S. C. Sep, Marijn Vellinga, R. Angela Sarabdjitsingh, Marian Joëls.

**Data curation:** Marijn Vellinga.

**Formal analysis:** Milou S. C. Sep.

**Funding acquisition:** Milou S. C. Sep, Marian Joëls.

**Investigation:** Milou S. C. Sep, Marijn Vellinga.

**Methodology:** Milou S. C. Sep, Marijn Vellinga, R. Angela Sarabdjitsingh.

**Project administration:** Milou S. C. Sep, R. Angela Sarabdjitsingh.

**Supervision:** Milou S. C. Sep, R. Angela Sarabdjitsingh, Marian Joëls.

**Visualization:** Milou S. C. Sep.

**Writing – original draft:** Milou S. C. Sep, Marijn Vellinga, Marian Joëls.

**Writing – review & editing:** R. Angela Sarabdjitsingh.

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
