## [Decision Letter · Decision Letter 0]

19 Apr 2021

PONE-D-21-07607

Measuring context-dependent memory in rodents: a systematic review and meta-analysis of important variables in the object-in-context task.

PLOS ONE

Dear Dr. Sep,

Thank you for submitting your manuscript to PLOS ONE. After careful consideration, we feel that it has merit but does not fully meet PLOS ONE’s publication criteria as it currently stands. Therefore, we invite you to submit a revised version of the manuscript that addresses the points raised during the review process.

Both reviewers liked the manuscript and suggested minor revisions which will improve the already high quality of the paper. Please address all comments and  submit your revised manuscript by Jun 03 2021 11:59PM. If you will need more time than this to complete your revisions, please reply to this message or contact the journal office at plosone@plos.org. Please include the following items when submitting your revised manuscript:

We look forward to receiving your revised manuscript.

Kind regards,

Patrizia Campolongo

Academic Editor

PLOS ONE

Journal Requirements:

Reviewers' comments:

Reviewer's Responses to Questions

**Comments to the Author**

1. Is the manuscript technically sound, and do the data support the conclusions?

Reviewer #1: Yes

Reviewer #2: Yes

2. Has the statistical analysis been performed appropriately and rigorously? 

Reviewer #1: Yes

Reviewer #2: Yes

3. Have the authors made all data underlying the findings in their manuscript fully available?

Reviewer #1: Yes

Reviewer #2: Yes

4. Is the manuscript presented in an intelligible fashion and written in standard English?

Reviewer #1: Yes

Reviewer #2: Yes

5. Review Comments to the Author

Reviewer #1: Sep et al. wrote an interesting systematic review and metanalysis on the object-in-context task in mice and rats, to identify critical variables, including strain, sex, experimental manipulations, differences in the behavioral paradigm, etc., that might influence the outcome of the task.

Based on their analyses the authors found that, among all variables examined, the sex, the scoring modality (manual vs automated), the size of the experimental arena, strongly influenced the behavioral outcome. The results obtained are very useful for informing future studies on context dependent memory performance.

Overall the metanalysis is interesting and well performed, and, in general, the methodology and analyses are sound and the selection of data included in the study is mostly clear.

My concern is related to the scarce information on the experimental group included in the study. Authors assert that naive, sham-operated or saline-injected rodents were included and that data were extracted taking into consideration the three types of control groups. However, for the saline-injected controls, for example, it would have been interesting to also consider as a variable the time of injection with respect to the behavioral paradigm (e.g. injection before the sample phase, injection before the test phase, etc.), as it is known that the injection itself might induce a stress response (Rao et al., 2012 Biol Psychiatry 72(6):466-75) which influences memory performance. Indeed authors in their metanalyses observed an inverted U-shaped dependency for prior arousal, with highest DR values in moderately aroused animal.

This limitation should be mentioned in the discussion.

Reviewer #2: This is a systematic review that provide an up-to-date overview and the protocol-related variations in the object-in-context task for rodents. The method and the data analysis used in this analytical review is undoubtedly adequate and knowing the factors that influence the performance of a task is important for the generalization of results between research groups. In this sense, the following suggestions would help to support this analytical review.

1. The Introduction needs a hypothesis. An analytical review also needs a hypothesis or at least a prognostic. In the Discussion you mention on two occasions that you already expected the observed results (p. 18), but in the Introduction, it is not mentioned why these results are expected.

2. Change “Methodological considerations section” from the Discussion section to the Method section, it is more appropriate in the Method section.

3. Include a likely explanation of how the highlighted factors affect behavior. In the Discussion these factors are only mentioned in a descriptive way. For example, sex and age. I understand that your objective was to look for factors that influence the behavior of the task, but that could already be inferred by just reviewing the literature. A tentative explanation would help for the originality of the manuscript and its justification.

4. I think you can omit from the title of the manuscript “Measuring context-dependent memory in rodents” since their work only represents results obtained from the analysis of “object-in-context task”.

5. Mention the factors that you identify that affect the task in the Abstract, instead of just mentioning that somewhere in the article you are going to give recommendations. I think this will help the reader to know if it is an article of interest.

6. PLOS authors have the option to publish the peer review history of their article (what does this mean?). If published, this will include your full peer review and any attached files.

Reviewer #1: No

Reviewer #2: **Yes: **Gina L. Quirarte

---

## [Author Response · Author response to Decision Letter 0]

1 Jun 2021

Responses to general comments

We have updated the manuscript according to these guidelines.

We have checked the references; no papers have been retracted. No changes to the references list were required.

Responses to Review Comments

Reviewer #1: 

Sep et al. wrote an interesting systematic review and metanalysis on the object-in-context task in mice and rats, to identify critical variables, including strain, sex, experimental manipulations, differences in the behavioral paradigm, etc., that might influence the outcome of the task.

Based on their analyses the authors found that, among all variables examined, the sex, the scoring modality (manual vs automated), the size of the experimental arena, strongly influenced the behavioral outcome. The results obtained are very useful for informing future studies on context dependent memory performance.

Overall the metanalysis is interesting and well performed, and, in general, the methodology and analyses are sound and the selection of data included in the study is mostly clear.

My concern is related to the scarce information on the experimental group included in the study. Authors assert that naive, sham-operated or saline-injected rodents were included and that data were extracted taking into consideration the three types of control groups. However, for the saline-injected controls, for example, it would have been interesting to also consider as a variable the time of injection with respect to the behavioral paradigm (e.g. injection before the sample phase, injection before the test phase, etc.), as it is known that the injection itself might induce a stress response (Rao et al., 2012 Biol Psychiatry 72(6):466-75) which influences memory performance. Indeed authors in their metanalyses observed an inverted U-shaped dependency for prior arousal, with highest DR values in moderately aroused animal.

This limitation should be mentioned in the discussion.

We agree with the reviewer that saline-injections before the sample and test phase could have differential effects, as it is known that arousal/stress can affect memory formation and retrieval differently [1].

As our primary aim was to summarize which variations in the object-in-context task affect its outcomes, we did not perform subgroup analysis (e.g. saline-injected animals alone), or focus on the sources of variation within each factor (e.g. time of injection within a control type). 

We aim to provide an overview that future studies can use to determine which variables need further exploration. Time of injection within saline-injected animals could be such a factor, although the relatively low variable importance of ‘control type’ (bottom 50%, figure 6) suggests that variation from this source has limited effects on overall performance.

We have addressed this point on p 19 of the discussion:

“Of note, it is also known that arousal and stress can affect memory formation and retrieval differently [1]. As a consequence, the time of saline-injection -which can trigger a mild stress response [2]- with respect to the OIC phases (e.g. before sample or test phase; or exactly when relatively to the test phase) could have caused variation in the saline-injected control animals, which was not accounted for in the current meta-analysis. However, type of control group ranked among the less important variables (bottom 50%), suggesting that variation from this source might have limited effects on overall performance.”

Reviewer #2: This is a systematic review that provide an up-to-date overview and the protocol-related variations in the object-in-context task for rodents. The method and the data analysis used in this analytical review is undoubtedly adequate and knowing the factors that influence the performance of a task is important for the generalization of results between research groups. In this sense, the following suggestions would help to support this analytical review.

1. The Introduction needs a hypothesis. An analytical review also needs a hypothesis or at least a prognostic. In the Discussion you mention on two occasions that you already expected the observed results (p. 18), but in the Introduction, it is not mentioned why these results are expected.

Our expectation was that the implementation and interpretation of the OIC would vary considerably across studies and that some of these variations would affect animals’ performance. 

We have added our expectation to the introduction (p5): “Considerable variation in OIC implementation among published studies is expected, and we expect that (some of) these variations affect animals’ performance.”

And adjusted the discussion accordingly (p15): “As expected, a substantial part (37.5%) of the variance could be explained by a set of moderators identified by a random forest approach, …”

We had no prior hypotheses about specific variables, and we agree with the reviewer that the use of ‘as expected’ in the previous version of the discussion was therefore incorrect. In the previous version of the manuscript ‘as expected’ was used to indicate that observations aligned with literature, not as reference to our a priori hypothesis. We have adjusted phrasing in the discussion accordingly:

(p18): “In line with frequently reported sex-dependence of spatial or neutral contextual memory formation across species [3,4], male animals showed a higher DR than females, but firm conclusions await more studies in females.”

(p19): “Smaller arenas resulted in higher DR values, as did large difference between context A and B.”

2. Change “Methodological considerations section” from the Discussion section to the Method section, it is more appropriate in the Method section.

We thank the reviewer for this suggestion. As the paragraph provides a reflection on the methodological strengths and limitations of the complete, performed analyses we feel it is best placed in the discussion. We agree that the section title can be confusing, and we have changed the title to “methodological strengths and limitations”.

3. Include a likely explanation of how the highlighted factors affect behavior. In the Discussion these factors are only mentioned in a descriptive way. For example, sex and age. I understand that your objective was to look for factors that influence the behavior of the task, but that could already be inferred by just reviewing the literature. A tentative explanation would help for the originality of the manuscript and its justification.

We agree with the reviewer that elaboration on these factors would benefit the manuscript. We have added a reflection on age:

(p 18): “ This is of interest, since context-dependent memory in humans shows a similar inverted U-shaped age-dependency [5]. Memory context-dependency increases with age in children, as they develop the ability to bind and integrate information [6]. As adults get older, context dependency decreases, which has been linked to age-related reductions in selective attention -leading to hyper-binding of too much contextual details thereby reducing accuracy for the relevant context- [7,8]; alterations in prefrontal-hippocampal connectivity [9]; and declines in hippocampal neurogenesis [10,11].”

And sex:

(p18): “In line with frequently reported sex-dependence of spatial or neutral contextual memory formation across species [3,4], male animals showed a higher DR than females, but firm conclusions await more studies in females. Performance in the OIC might be enhanced in males compared to females, as the former are more likely to adopt a hippocampus-dependent place-strategy to navigate an environment (i.e. navigate based on a contextual map) than the latter (who are more likely to use a striatum-dependent landmark strategy) [12]. Such place-strategies could create a stronger representation of the OIC context in the brain, enabling richer context-dependent memories.”

4. I think you can omit from the title of the manuscript “Measuring context-dependent memory in rodents” since their work only represents results obtained from the analysis of “object-in-context task”.

We agree with the reviewer, we have changed the title to “The rodent object-in-context task: a systematic review and meta-analysis of important variables”. We have also changed the short title to “Meta-analysis of the rodent object-in-context task”

5. Mention the factors that you identify that affect the task in the Abstract, instead of just mentioning that somewhere in the article you are going to give recommendations. I think this will help the reader to know if it is an article of interest.

We have added these variables to the abstract: “Based on these observations we provide recommendations on sex, strain, prior arousal, context (size, walls, shape, etc.) and timing (habituation, learning, and memory phase) to create more consensus in the set-up, procedure and interpretation of the object-in-context task for rodents” 

References

1. Quaedflieg CWEM, Schwabe L. Memory dynamics under stress. Memory. 2018;26: 364–376. doi:10.1080/09658211.2017.1338299

2. Rao RP, Anilkumar S, Mcewen BS, Chattarji S. Glucocorticoids Protect Against the Delayed Behavioral and Cellular Effects of Acute Stress on the Amygdala. BPS. 2012;72: 466–475. doi:10.1016/j.biopsych.2012.04.008

3. Andreano JM, Cahill L. Sex influences on the neurobiology of learning and memory. Learn Mem. 2009;16: 248–266. doi:10.1101/lm.918309

4. Keeley RJ, Bye C, Trow J, Mcdonald RJ. Strain and sex differences in brain and behaviour of adult rats: Learning and memory, anxiety and volumetric estimates. Behav Brain Res. 2015;288: 118–131. doi:10.1016/j.bbr.2014.10.039

5. Sep MSC, Joëls M, Geuze E. Individual differences in the encoding of contextual details following acute stress: An explorative study. Eur J Neurosci. 2020; ejn.15067. doi:10.1111/ejn.15067

6. Imuta K, Scarf D, Carson S, Hayne H. Children’s learning and memory of an interactive science lesson: Does the context matter? Dev Psychol. 2018;54: 1029–1037. doi:10.1037/dev0000487

7. Powell PS, Strunk J, James T, Polyn SM, Duarte A. Decoding selective attention to context memory: An aging study. Neuroimage. 2018;181: 95–107. doi:10.1016/j.neuroimage.2018.06.085

8. Strunk J, James T, Arndt J, Duarte A. Age-related changes in neural oscillations supporting context memory retrieval. Cortex. 2017;91: 40–55. doi:10.1016/j.cortex.2017.01.020

9. Ankudowich E, Pasvanis S, Rajah MN. Age-related differences in prefrontal-hippocampal connectivity are associated with reduced spatial context memory. Psychol Aging. 2019;34: 251–261. doi:10.1037/pag0000310

10. Kirschen GW, Ge S. Young at heart: Insights into hippocampal neurogenesis in the aged brain. Behav Brain Res. 2019;369: 111934. doi:10.1016/j.bbr.2019.111934

11. Alam MJ, Kitamura T, Saitoh Y, Ohkawa N, Kondo T, Inokuchi K. Adult neurogenesis conserves hippocampal memory capacity. J Neurosci. 2018;38: 6854–6863. doi:10.1523/JNEUROSCI.2976-17.2018

12. Yagi S, Galea LAM. Sex differences in hippocampal cognition and neurogenesis. Neuropsychopharmacology. 2019;44: 200–213. doi:10.1038/s41386-018-0208-4

---

## [Editor Report · Decision Letter 1]

25 Jun 2021

The rodent object-in-context task: a systematic review and meta-analysis of important variables

PONE-D-21-07607R1

Dear Dr. Sep,

We’re pleased to inform you that your manuscript has been judged scientifically suitable for publication and will be formally accepted for publication once it meets all outstanding technical requirements.

Kind regards,

Patrizia Campolongo

Academic Editor

PLOS ONE
---

## [Editor Report · Acceptance letter]

8 Jul 2021

PONE-D-21-07607R1 

The rodent object-in-context task: a systematic review and meta-analysis of important variables 

Dear Dr. Sep:

I'm pleased to inform you that your manuscript has been deemed suitable for publication in PLOS ONE. Congratulations! Your manuscript is now with our production department. 

Kind regards, 

on behalf of

Dr. Patrizia Campolongo 

Academic Editor

PLOS ONE